# Bolstering CD8^+^ T Cells’ Antitumor Immunity: A Promising Strategy to Improve the Response to Advanced Prostate Cancer Treatment

**DOI:** 10.3390/biology14050544

**Published:** 2025-05-14

**Authors:** Beijing Dang, Lixin Liang, Zhijun Li, Junli Luo, Shangwei Zhong

**Affiliations:** The Cancer Research Institute, Hengyang Medical School, University of South China, Hengyang 421009, China; 2022201311122@stu.usc.edu.cn (B.D.); 20215740119@stu.usc.edu.cn (L.L.); 2010020037@usc.edu.cn (Z.L.); jlluo@usc.edu.cn (J.L.)

**Keywords:** CD8^+^ T cells, prostate cancer, the immune-suppressive microenvironment, immunotherapy

## Abstract

Prostate cancer, typically considered as an immunologically “cold” tumor due to its sparse immune infiltration and low immunogenicity. However, accumulating evidences show that CD8⁺ T cells are closely associated with both prostate cancer progression and its responses to therapy. Recent insights into CD8⁺ T cell dynamics facilitate the identification of new molecular targets and the development of innovative immunotherapeutic strategies, offering a promise for more effective treatment of advanced castration resistant prostate cancer.

## 1. Introduction

Prostate cancer (PCa) is one of the leading causes of cancer-related deaths among men in the Western world and its incidence rate continues to rise in Asia [1,2,3,4]. The main therapeutic strategies for patients, particularly those with the advanced disease, include androgen-deprivation therapy (ADT), immunotherapy, gene therapy, radiotherapy, and chemotherapy [5,6,7,8,9,10,11,12]. Although these treatments provide significant benefits, management remains challenging [8]. For instance, ADT is a standard therapy for the advanced prostate cancer, and while patients initially respond well to it, most eventually progress to the castration-resistant (CRPC) stage, a highly aggressive form of the disease with limited treatment options [9].

Significant advances have been made in understanding the molecular characterization of the localized, recurrent, and progressive disease, as well as the underlying mechanisms of progression and therapeutic resistance [13,14,15,16,17,18,19]. Research indicates that progression is closely associated with androgen receptor (AR) signaling, loss of tumor suppressor genes, growth factor signaling, metabolic changes, the immune system, and the tumor microenvironment (TME) [13,14,20,21,22,23,24]. Several reviews have summarized the roles of genetic pathways in development and therapeutic resistance [13,25,26,27].

The TME is a crucial component of tumor tissue, characterized by hypoxia and acidity, and consists of both cellular and noncellular components [28,29,30]. The cellular components include various stromal cells such as endothelial cells (ECs) and cancer-associated fibroblasts (CAFs), CD8^+^ T cells, myeloid-derived suppressor cells (MDSCs), tumor-associated macrophages (TAMs), natural killer (NK) cells, neutrophil, and regulatory T cells (Tregs). The noncellular components comprise the extracellular matrix (ECM), cytokines, chemokines, growth factors, and various metabolic products. Both cellular and noncellular components are closely linked to the onset, progression, and metastasis of [30,31,32,33,34,35].

Immunosuppression is one of the most critical characteristics of the TME in [36,37,38]. In this immunosuppressive environment, the functions of CD8^+^ T cells and NK cells are significantly inhibited, which is a major reason why prostate cancer cells can evade immune surveillance [39,40,41,42]. The status of CD8^+^ T cells varies with cancer progression. For example, higher levels of CD8^+^ T cell exhaustion have been observed in high-grade prostate cancer (HG) compared to low-grade (LG) [43]. The NK cells play an important role by directly killing tumor cells. However, in the TME, immune-suppressive factors, such as TGF-β, IL-10, and immune checkpoint molecules, can inhibit the function of NK cells, interrupting their antitumor activity [44]. The neutrophils can promote tumor progression and metastasis by enhancing inflammation, secreting proteases, and supporting angiogenesis. Meanwhile, neutrophils can suppress the function of other immune cells through the secretion of immunosuppressive factors, thereby facilitating tumor immune evasion [45,46].

The formation of the immunosuppressive TME is attributed to immunosuppressive stromal cells and factors derived from these and from the stromal cells [47]. Tregs can inhibit the proliferation and activation of effector T cells directly or through secreted factors such as IL-10, IL-4, and TGF-β. MDSCs promote tumor immune escape by secreting factors like IL-10 and TNF-β, downregulating the TCR-related ζ chain, or recruiting Tregs to inhibit T cell activation. Immunosuppressive factors include various cytokines and chemokines such as IL-10, TGF-β, and TNF-β [22,30,35,48]. Abnormal gene expressions in cells can significantly modulate immune microenvironment in tumors [48,49,50,51,52]. For example, in prostate cancer cells, upregulation of nuclear cap-binding protein 2 (NCBP2) can enhance the expression of immunosuppressive factors (such as TGF-β and IL-10) or induce the production of chemokines (such as CCL22 and CCL2), which is closely associated with a decrease in CD8^+^ T cells and an increase in Treg cells and neutrophils at the tumor site, thereby creating an immunosuppressive microenvironment [48]. High expression of N-acetyltransferase 10 (NAT10) in cells can suppress CD8^+^ T cell recruitment and cytotoxicity via the CCL25/CCR9 axis to form an immunosuppressive microenvironment [49].

Recent research has revealed the essential roles of CD8^+^ T cells in the progression of [50,51,52,53,54]. Concurrently, a range of combinatorial immunotherapies have been developed to reverse or reactivate the immunosuppressed CD8^+^ T cells in the TME, restore CD8^+^ T cell infiltration, modify the genes of a patient’s own T cells (CAR T cell therapy), and target immunosuppressive stromal cells or factors in tumors. Some of these strategies have demonstrated promising therapeutic effects in preclinical models and clinical trials, offering potential new options for treatment [55,56,57,58,59,60,61,62]. The application of CAR T cell therapy in treatment has been well documented [63,64,65,66,67,68]. Meanwhile, bispecific T cell engager (BiTE) is another important T cell therapy. For example, AMG509, also known as xaluritamig, is a bispecific antibody that can simultaneously bind to two different targets, one on the T cells and the other on the cancer cells. AMG509 binds to PSMA on the surface of prostate cancer cells, specifically targeting the cancer cells. At the same time, it binds to the CD3 receptor on T cells, activating them and prompting them to release cytotoxic molecules to kill the cancer cells. This therapy holds significant clinical importance in the treatment of advanced prostate cancer, including castration-resistant prostate cancer [69,70].

In this review, we focus on recent advances in understanding the roles of CD8^+^ T cells in development and therapeutic resistance, as well as new strategies to reverse the immunosuppressive status of CD8^+^ T cells or increase their infiltration to improve the efficacy of advanced treatment.

## 2. Heterogeneity of CD8^+^ T Cells in PCa

### 2.1. Evolution and Development of CD8^+^ T Cells

The maturation of CD8⁺ T cells begins when bone marrow–derived precursors migrate to the thymus, where they undergo TCR β-chain rearrangement and pre-TCR formation during the double-negative (DN) stage, then proceed to the double-positive (DP) stage for α-chain rearrangement. In the thymic cortex, DP cells whose TCRs bind self-peptide–MHC I complexes with moderate affinity are “positively selected” and differentiate into CD8 single-positive cells; subsequently, in the thymic medulla, cells that bind self-antigens with high affinity are eliminated through “negative selection”. The surviving mature CD8⁺ T cells then exit the thymus and enter peripheral tissues to perform antigen-specific, cytotoxic immune functions (Figure 1A) [71].

### 2.2. Dysfunction of CD8^+^ T Cells in PCa

Peripheral blood and nearby lymph nodes are the primary sources of the CD8^+^ T cells that infiltrate prostate cancer tissues, and there is a demonstrable correlation between the immune profiles of the peripheral blood and those within the tumor microenvironment [71,72,73,74]. Tumors affect the composition of peripheral blood by altering immune cell function, secreting immune-suppressive factors, and releasing specific markers, providing important information for tumor diagnosis, monitoring, and prognosis assessment [71]. In addition, the detection of circulating tumor cells (CTCs) can reveal the presence of tumor cells in the blood, and the number of CTCs is closely related to tumor metastasis and prognosis [72]. Studies have shown that an increase in CTCs usually indicates a higher risk of tumor metastasis [73]. At the same time, some cellular components of TME secrete immune-suppressive factors such as TGF-β and IL-10, which inhibit the antitumor activity of immune cells in peripheral blood, helping tumor cells evade surveillance by the host immune system [74].

CD8^+^ T cells are crucial antitumor effector cells capable of infiltrating tumor tissue. Once activated by tumor antigens recognized in the context of MHC class I molecules, CD8^+^ T cells differentiate into effector cells with potent antitumor activity [75,76]. However, within the prostate tumor site, CD8^+^ T cells often fail to become fully activated or become exhausted due to chronic and sustained stimulation by tumor antigens and various immunosuppressive factors in the TME (Figure 1B). This results in reduced proliferation and increased apoptosis of the CD8^+^ T cells, along with disrupted secretion of IFN-γ, TNF-α, granzyme B (GZMB), and perforin (PFP). This state of the CD8^+^ T cells within the tumor tissue is known as “T cell exhaustion” [77,78]. CD8^+^ T cell dysfunction within prostate cancer tissues represents a critical mechanism driving the marked impairment of CD8^+^ T cells’ antitumor activity, and the CD8^+^ T cell dysfunction is closely linked to disease progression [55,79].

### 2.3. Prognostic Value of CD8^+^ T Cell Infiltration of Prostate Tumors

PCa is often described as an immune “cold” tumor due to the relatively low number of immune cells in the TME and the poor immunogenic response. However, T cell infiltration in prostate tumors has been observed for decades [80,81,82,83,84,85,86]. For example, a mouse model of PCa liver metastasis showed that CD8^+^ T cells are very rare in liver metastatic tissue [87]. Similarly, limited infiltration of CD8^+^ T cells was found in prostate tumors in black African male patients [85]. However, Ness and colleagues found a high density of CD8^+^ T cells in prostate tumor tissue and demonstrated that this high density is an independent negative prognostic marker of biochemical failure-free survival (BFFS) for PCa [54]. Another study indicated that, for node-positive PCa patients, a high density of CD8^+^ T cells in tumors and the expression of PD-L1 in cancer cells may indicate a higher risk of clinical progression [87]. In contrast, Goulielmaki and colleagues found that a high density of HER-2/neu(780–788)-specific CD8^+^ T cells in the peripheral blood of PCa patients undergoing conventional treatment was associated with better progression-free survival (PFS) compared to patients with a low density of the cells [88]. For patients undergoing radical prostatectomy (RP), high intra-tumoral CD8^+^ T cell infiltration is independently associated with improved overall survival [89]. Additionally, during PCa treatment, the density of T cells in the tumor site can vary. ADT has been shown to induce increased infiltration of CD4^+^ T cells, CD8^+^ T cells, and Treg cells into tumor sites in human prostates, suggesting that androgen deprivation may affect PCa immunotherapy [80,81]. Cryoablation also shows local immune modulation in tumor tissue, with significant increases in tumor-infiltrating CD8^+^ T cells after PCa cryoablation [81]. These contradictory reports may be due to the complex immune microenvironment of PCa.

PCa generally exhibits a “cold” immune microenvironment, with low basal CD8^+^ T-cell infiltration in many settings, however, both the density and functional competence of these T cells can be reshaped by tumor stage, antigen specificity, and therapeutic interventions. These factors together determine whether CD8^+^ T-cell presence confers a favorable or adverse prognosis. Future studies must elucidate the signals that promote or inhibit CD8^+^ T-cell recruitment across tumor stages. More research is needed to define how antigen specificity and T-cell receptor (TCR) clonotype dynamics and the metabolic constraints of the “cold” microenvironment limit their function. Then it may be possible to identify the optimal treatment windows and combinations (e.g., androgen deprivation, radiotherapy, immune-checkpoint blockade, and microbiome modulation) to enhance T-cell infiltration. In parallel, researchers need to develop biomarkers that predict and monitor CD8^+^ T-cell responses in real time, and dissect the immunosuppressive networks that sustain tumor “coldness”; and the ultimate goal is to convert prostate cancer into an immunologically “hot” state.

### 2.4. New CD8^+^ T-Cell Subsets in Prostate Cancer Identified by Single-Cell Omics

The application of single-cell RNA sequencing (sc RNA sequence) has recently provided unprecedented opportunities to assess thousands of cells at the single-cell level within a sample and establish the immune cellular landscape of human prostate tumor tissues (Figure 1C). Using sc RNA sequence, a study revealed that the transcription of various innate and adaptive immune cells was perturbed in PCa compared to normal prostate tissue at the single-cell level. The expression levels of exhaustion-associated genes were significantly increased in CD8^+^ T cells, while the expression of immune-recruiting and activating chemokines and cytokines by fibroblast and epithelial cells was reduced in PCa [90]. Consistently, expressions of dysfunctional markers (PDCD1 and LAG3) were increased, while expressions of effector cytokines (IFNG) were decreased in CD8^+^ T cells within the cribriform PCa TME [91].

Thanks to single-cell omics, new CD8^+^ T-cell subsets have been identified, enhancing our understanding of the heterogeneity of CD8^+^ T cells in PCa [92] (Figure 1C and Table 1). Chen and colleagues found that infiltrating CD8^+^ T effector cells exhibit substantial heterogeneity, comprising three clusters (2, 3, and 5). Cluster 5 is the least activated subset due to its lowest levels of metabolism and immune pathways (Figure 2A), while cluster 3 is a highly activated subtype. Interestingly, PCa cells can alter the transcriptome of infiltrating CD8^+^ effector T cells to express tumor marker genes such as FOLH1, mediated by cancer cell-derived extracellular vesicles (EVs). Additionally, cancer-derived EVs can induce ectopic KLK3 expression in infiltrating T cells. High KLK3 expression in T cells is associated with the micro-metastases of PCa, and further studies suggest that the immune cell transcriptome may be altered in lymph nodes (LNs) to establish a pre-metastatic niche before actual metastasis occurs (Figure 2B,F) [93].

Single-cell proteomics revealed that the immune landscape is vastly different between prostate tumors and benign adjacent tissue. Five different T-cell phenotypes were identified, with two T-cell subsets (TC03 and TC04) being significantly enriched in prostate tumors compared to adjacent samples (Figure 2C) [94]. By combining single-cell and spatial transcriptomic analyses, the immune-suppressive microenvironment of PCa tumors was dissected. The study showed that the immune-suppressive TME is attributed to suppressive myeloid populations, exhausted T cells, and high stromal angiogenic activity. Three subsets of CD8^+^ T cells were identified: CTL-1, CTL-2, and CD8^+^ effector T cells (Figure 2D,E). The T-cell cytotoxicity score of these three CD8^+^ T-cell subsets is much lower than that of T cells in “hot” tumors, such as head and neck squamous cell carcinoma, lung cancer, and liver hepatocellular carcinoma. Additionally, the CTL-1 and CD8^+^ effector T cells in prostate tumors are dysfunctional, with high expression of T-cell exhaustion genes, and show much higher exhaustion scores compared to healthy prostate tissues. Interestingly, the T-cell exhaustion score is also higher in tumor-adjacent normal samples than in healthy prostate tissues, suggesting that the immune microenvironment in tumor-adjacent regions has already been altered due to the impact of nearby tumor tissues [31]. An exhausted cytotoxic CD8^+^ T-cell subpopulation was also observed at bone metastasis sites in human metastatic PCa [39].

## 3. Factors and Mechanisms for the Dysfunction of CD8^+^ T Cells in PCa

The immune-suppressive microenvironment is a key factor in PCa progression to CRPC, which typically shows a poor response to immunotherapy [96,97,98]. CD8^+^ T-cell dysfunction is a major event in the immune suppression observed in PCa tumors. Understanding the mechanisms underlying this dysfunction can enhance our comprehension of PCa progression and aid in the development of new therapeutic strategies. CD8^+^ T-cell dysfunction in tumors is attributed to various factors, including suppressive myeloid populations (MDSCs), Treg cells, TAMs, and immune-suppression-related noncellular components in the TME (Figure 3) [22,48].

### 3.1. Cellular Components in Tumors Contribute to CD8^+^ T-Cell Dysfunction

MDSCs, Treg cells, and M2-type TAMs are primarily immunosuppressive cellular components in the TME. In PCa bone metastases, a TAM population with M2 macrophage characteristics was identified. This subset suppresses CD8^+^ T-cell activation through the expression of epidermal growth factor-like ligands and the mediation of the JAK/STAT3 pathway, promoting PCa tumor growth [39]. Another TAMs subset, Spp1hi-TAMs, can suppress CD8^+^ T-cell activity and promote immune checkpoint inhibitors (ICIs) resistance. Targeting adenosine A2A receptors (A2ARs) significantly decreases SPP1hi-TAM abundance in CRPC, and enhances the CRPC responsiveness for PD-1 blockade [99]. Some studies indicate that androgen ablation can temporarily increase CD8^+^ T-cell infiltration and enhance effector function in PCa. However, this effect is transient as Treg cells subsequently expand, countering the increased CD8^+^ T-cell function. Treg-cell expansion and maintenance are regulated by IL-2, and blocking IL-2 with an IL-2-neutralizing antibody can rescue CD8^+^ T-cell effector responses by significantly reducing the percentage and number of Treg cells [100].

Our study demonstrated that during the shrinking stage of PCa tumors after ADT, cancer cell-derived odorant-binding protein (OBP2A) interacts with CXCL15/IL8 to recruit MDSCs into the TME (Figure 3B). These infiltrated MDSCs significantly suppress CD8^+^ T-cell proliferation in tumor tissue, promoting androgen-independent PCa cell growth and leading to the onset of castration resistance [14]. Furthermore, in PCa tissue, MDSCs (Gr1^+^) inversely correlate with infiltrated CD8^+^ T cells. Targeting MDSCs with multikinase inhibitors such as cabozantinib and BEZ235, combined with immune checkpoint blockade, shows robust synergistic responses in primary and metastatic CRPC. One underlying mechanism is that targeting MDSCs alleviates the immune suppression of CD8^+^ T cells, thereby enabling them to attack PCa cells [101]. Xu and colleagues found that enzalutamide treatment for CRPC can increase monocytic myeloid-derived suppressor cell (M-MDSC) populations and PD-L1 expression while decreasing CD8^+^ T-cell numbers, resulting in enzalutamide resistance [102,103]. Wang and colleagues found that the administration of white button mushroom (WBM) suppresses polymorphonuclear MDSCs (PMN-MDSCs) by interrupting STAT3/IRF1 and TGFβ signaling, along with causing an increase in CD8^+^ T cells and NK cells [104].

Importantly, MDSCs can crosstalk with Treg cells to establish a T-cell suppression and pro-tumor TME in PCa. A group of ligand-receptor interactions between these cell populations has been identified. A representative axis of interaction is CCL20–CCR6, with CCL20 expressed in terms of MDSCs and CCR6 expressed in terms of Treg cells (Figure 3D). The CCL20–CCR6 axis plays a crucial role in the prostate immune-suppressive TME. A CCL20-blocking antibody can significantly reduce prostate tumor growth in a mouse model by interrupting the CCL20–CCR6 interaction (Figure 3C,D) [31]. Crosstalk between macrophages and CD8^+^ T cells is also involved in suppressing CD8^+^ T-cell function. A study showed that UBC9-mediated STAT4 SUMOylation facilitates the proinflammatory activation of macrophages (Figure 3E). Treatment with the UBC9 inhibitor 2-D08 promoted the antitumor effect of TAMs while increasing PD-1 expression in CD8^+^ T cells. Therefore, the UBC9 inhibitor exhibited synergistic antitumor efficacy when combined with immune checkpoint blockade therapy [105]. Other stromal cells also affect CD8^+^ T-cell density in PCa tumors. S100A11, a member of the S100 protein family with calcium ion-binding capabilities, is involved in various cellular processes, including proliferation, differentiation, apoptosis, and tumor formation. In gastric cancer, S100A11 promotes cell proliferation and invasion by interacting with the NF-κB signaling pathway (Figure 3A) [106]. In prostate cancer, Han and colleagues found that Erdafitinib treatment combined with S100A11 knockdown in PCa cells and CAFs increased infiltration of effective CD8^+^ T cells in the tumor, thereby reducing tumorigenicity (Figure 3A) [107].

### 3.2. Noncellular Components in Tumors Contribute to CD8^+^ T-Cell Dysfunction

In addition to immunosuppressive cellular components, noncellular components in tumors contribute to CD8^+^ T-cell dysfunction in PCa tumors. FasL-expressing exosomes derived from PCa cells inhibit antitumor T-cell responses by inducing CD8^+^ T-cell apoptosis, facilitating tumor immune evasion (Figure 3A) [108]. Interestingly, highly malignant PCa cells, such as DU145 cells, secrete exosomes expressing higher levels of programmed necrosis ligand 1 (PD-L1) (Figure 3A). PCa cells like LNCaP, with low levels of PD-L1 expression, can uptake exosomal PD-L1 to inhibit CD8^+^ T-cell function, thereby escaping CD8^+^ T-cell-mediated killing [54]. Furthermore, PCa cells can transport interleukin-8 (IL-8) to CD8^+^ T cells through exosomes, activating PPARα in CD8^+^ T cells. This results in decreased glucose utilization and increased fatty acid catabolism. PPARα further activates uncoupling protein 1 (UCP1) to interfere with energy metabolism. Consequently, the tumor exosome-activated IL-8-PPARα-UCP1 axis induces CD8^+^ T-cell exhaustion and fosters PCa cell immune evasion (Figure 3F) [56].

Additionally, genes in PCa cells can affect CD8^+^ T-cell infiltration and function. CD8^+^ T-cell infiltration can be regulated by SEPTIN5 (SEPT5) in PCa. Downregulation of SEPT5 can significantly increase CD8^+^ T-cell infiltration due to the elevated expression of CCL5, CXCL5, CXCL9, CXCL10, and IFNGR1, which are associated with immune cell (CD8^+^ T cell) infiltration [109]. Zhu and colleagues found that the chromatin effector Pygopus 2 (PYGO2), a driver oncogene, orchestrated a p53/Sp1/Kit/Ido1 signaling network to form an immunosuppressive TME, promoting PCa progression (Figure 3A). Pygo2 inhibition augmented the activation and infiltration of CD8^+^ T cells and enhanced the antitumor efficacy of immunotherapies such as immune checkpoint blockade (ICB) [57]. Meanwhile, signaling pathways in CD8^+^ T cells also contribute to their dysfunction in PCa tumors. The AR in CD8^+^ T cells can repress IFNγ expression, resulting in T-cell dysfunction. AR inhibition can prevent T-cell exhaustion and improve checkpoint blockade efficacy in PCa treatment [103].

## 4. Strategies for Improving CD8^+^ T-Cell Function in PCa Treatment

### 4.1. Autologous Cellular Immunotherapy in PCa

Sipuleucel-T is an autologous cellular immunotherapy used to treat metastatic castration-resistant prostate cancer (mCRPC), which is defined as prostate cancer that, despite maintaining serum testosterone at castration levels, exhibits biochemical or radiographic progression and has developed distant metastases. It works by stimulating the patient’s own immune components to recognize and attack prostate cancer cells. The therapy involves isolating dendritic cells (DCs) from the patient’s blood, and then the DCs are incubated in the laboratory with the prostate cancer–associated antigen PAP-GM-CSF. This process activates the DCs, which are then re-infused into the patient’s body. The activated immune system cells stimulate a targeted immune response against prostate cancer cells expressing the PAP protein. Sipuleucel-T is considered a form of cancer vaccine therapy that helps to enhance the immune system’s ability to attack cancer, although it is not a conventional vaccine like those used to prevent infections [110]. It is approved for patients with prostate cancer who have no or limited symptoms but are not responding to hormone therapy. While the therapy has been shown to extend survival in some prostate cancer patients, its effectiveness can vary, and it may be associated with side effects such as fever, chills, fatigue, and other immune-related responses [111].

### 4.2. The Microbiome and CD8^+^ T Cells in PCa

Given the relatively immunological ‘‘cold’’ characteristics of PCa, an effective strategy to enhance PCa treatment, especially for CRPC, is to convert PCa tumors from “cold” to “hot”. This involves increasing tumor-infiltrating T cells and improving antitumor responses [33,96,112]. Luo and colleagues found that Akkermansia muciniphila-derived extracellular vesicles (Akk-EVs) can increase the proportion of GZMB^+^CD8^+^ and IFN-γ^+^CD8^+^ T cells and M1-like macrophages while reducing M2-like macrophages, so demonstrating good efficacy in PCa treatment [113].

Emerging evidence indicates that both gut and tumor-resident microbiota can profoundly shape CD8⁺ T-cell responses in prostate cancer. For instance, Xu et al. showed that the botanical combination icaritin-curcumol remodels the gut microbiome—boosting short-chain fatty acid production—and thereby enhances CD8⁺ T-cell infiltration, proliferation (Ki67⁺), and cytotoxic effector functions (perforin, granzyme A/B, IFN-γ) in prostate tumors [60]. More broadly, commensal bacteria have been found to translocate into tumor niches where they “educate” CD8⁺ T cells, tuning their activation threshold and metabolic programs to control tumor growth. Altered microbial metabolites—such as butyrate or tryptophan derivatives—can also modulate the CD8⁺ T-cell glucose versus fatty-acid oxidation balance, impacting their persistence and reversal of exhaustion within the tumor microenvironment [114,115].

### 4.3. ICB Therapy in PCa

Several therapeutic delivery systems show a synergistic antitumor effect when combined with immune checkpoint blockade (ICB) due to augmented CD8^+^ T-cell infiltration or reactivation in tumors (Table 2). Vardeu and colleagues developed a prostate-specific four-antigen cassette with a viral vector, chimpanzee adenovirus Oxford 1 (ChAdOx1), expressing prostate-specific antigen (PSA). The prime-boost regimen induced high-magnitude, functional CD8^+^ T cells in murine PCa models through intravenous administration [116]. Another strategy is that irreversible electroporation (IRE) combined with anti-CTLA-4 ICB therapy significantly improved tumor treatment outcomes compared to anti-CTLA-4 monotherapy or IRE alone, as the combined therapy augmented the expansion of tumor-specific CD8^+^ T cells in tumors and blood. Importantly, this strategy promoted the establishment of tumor antigen-specific tissue-resident memory CD8^+^ T cells, providing protection from secondary tumor challenges [117]. Although CTLA-4 is a good therapeutic target, CTLA-4 is found to be an inhibitory receptor that plays a key role in suppressing T-cell activation, particularly in Tregs. While CTLA-4 blockade is used to enhance immune responses in cancer therapy, it can also promote Treg expansion, potentially inhibiting antitumor immunity [118]. Additionally, this blockade may contribute to immune-related adverse events (irAEs), such as intestinal inflammation [119]. Therefore, the effects of CTLA-4 blockade on Tregs need to be carefully evaluated to optimize therapeutic benefits and reduce side effects. Moreover, one Phase II clinical study, using longitudinal single-cell sequencing, demonstrates that androgen-deprivation therapy combined with anti–PD-1 immunotherapy significantly enhances immune cell infiltration—particularly CD8⁺ T cells—in metastatic castration-sensitive prostate cancer lesions [98].

In addition, an anti-PSCA modular system (antibody) can trigger efficient T-cell-mediated targeting of PSCA^+^ PCa cells by activating both CD4^+^ and CD8^+^ T cells, offering a new therapeutic option for PCa treatment [120]. Furthermore, selective YAP1 depletion in ECM–associated CAFs (ECM-CAF) can promote ECM-CAF switching to lymphocyte-associated CAFs (Lym-CAF), which is an antitumor CAF subset, and enhanced the infiltration and activation of CD8^+^ T cells. Thus, the specific targeting of YAP1 in ECM-CAF can switch the pro-tumorigenic feature of CAFs to antitumor phenotypes in PCa; and combining selective YAP1 depletion with anti-PD-1 antibodies can enhance the immunotherapeutic effect [121]. Importantly, Shenderov et al. conducted a single-arm phase II clinical trial to evaluate the safety and efficacy of Enoblituzumab as a neoadjuvant therapy. Enoblituzumab enhances the antitumor functions of T cells and NK cells by targeting B7-H3. It not only activates T cells and NK cells, enhancing their ability to kill tumor cells, but also helps to reverse T-cell exhaustion and restore their normal function. Additionally, Enoblituzumab reduces the immunosuppressive effects within the TME by inhibiting immune-suppressive cells such as Tregs and M2 macrophages, thereby enhancing the overall immune response. The study found that among patients treated with Enoblituzumab, 66% had undetectable PSA levels 12 months after surgery, indicating that the treatment may effectively reduce tumor activity and size, thus improving patient cure rates [122].

### 4.4. Chemotherapy Therapy in PCa

A group of chemical compounds can increase CD8^+^ T-cell infiltration and improve antitumor responses in PCa when combined with immunotherapy. Zoledronic acid (ZA), a treatment for osteoporosis, combined with peptide/polyinosinic-polycytidylic acid (poly-IC) vaccination, showed a synergistic effect and induced a high antigen-specific CD8^+^ T-cell response, reducing prostate tumor growth and prolonging survival. ZA plus thymosin α1 (Tα1) treatment increased CD8^+^ T-cell infiltration and enhanced tumor inflammation by activating the MyD88/NF-κB pathway in macrophages and T cells in PCa tumors, consequently suppressing PCa progression [123]. P21-activated kinase-4 (PAK4) inhibitors, PF-3758309 (PF) and KPT-9274 (KPT), improved the response of PCa to PD1 antibody (αPD1) therapy by significantly increasing CD8^+^ T-cell infiltration and IFNγ expression in tumors [124].

Overall, in standard mCRPC patients, the combination of docetaxel and anti–PD-1 has not changed the standard of care [125], however, in small-cell/neuroendocrine subtypes, and when paired with novel approaches such as tumor vaccines, it has shown potential synergistic benefits [97,126] (Table 2). Future efforts must focus on more precise patient-selection biomarkers and on optimizing treatment sequencing and combinations to truly harness the advantages of immunotherapy in prostate cancer.

### 4.5. CAR T-Cell Therapies and Vaccines in PCa

CAR-T therapy is a precision immunotherapy that genetically modifies a patient’s own T cells to express specific chimeric antigen receptors, thereby directing them to recognize and kill tumor cells. Additionally, various vaccines and CAR T-cell therapies have been developed for PCa treatment (Table 2). Peptide vaccines represent a promising immunotherapeutic approach, with their efficacy in tumor inhibition partially attributed to the induction of augmented CD8^+^ T-cell antitumor responses [127,128,129,130,131]. Building on vaccine-based approaches, Obradovic and colleagues combined degarelix with a granulocyte-macrophage colony-stimulating factor (GM-CSF)-secreting allogeneic cellular vaccine plus low-dose cyclophosphamide (Cy/GVAX). The degarelix plus Cy/GVAX combination generated a robust T-cell response against PCa, demonstrating a significant treatment effect (Table 2) [132]. A DNA vaccine encoding prostatic acid phosphatase (MVI-816), combined with pembrolizumab, was tested in a clinical trial for patients with metastatic CRPC. The combined treatment resulted in a longer time to progression due to irAEs with augmented plasma cytokines associated with immune activation and CD8^+^ T-cell recruitment [59].

CAR T-cell therapy has shown potent efficacy in some cancers, and PCa-specific CAR T-cell therapies have been developed (Table 2) [57,70]. Bhatia and colleagues developed a STEAP1-directed CAR T-cell therapy based on the broad expression of six transmembrane epithelial antigens of the prostate 1 (STEAP1) in lethal metastatic PCa. To evade STEAP1 antigen escape, they combined STEAP1 CAR T cells with tumor-localized IL-12 immunotherapy. This combination therapy remodeled the immune TME of PCa and significantly enhanced antitumor efficacy (Table 2) [133]. Ajmal and colleagues found that adrenoceptor beta-2 (ADRB2) knockdown in CAR T cells can enhance its antitumor responses via the ZAP-70/NF-κB signaling axis, and their developed ADRB2 knockdown CAR T (shβ2-CAR T) cells showed better efficacy against PCa than conventional CAR T cells [134].

Another ideal target for CAR T-cell therapy is prostate-specific membrane antigen (PSMA), which is a protein highly expressed on the surface of prostate cancer cells. Therefore, T cells can be engineered to specifically recognize PSMA on cancer cells (Table 2). One of the challenges faced by CAR T-cell therapy is the immunosuppressive TME, where cytokines like TGF-β can inhibit T-cell function. To overcome this issue, Narayan et al. modified the CAR T cells to make them insensitive to TGF-β signaling, allowing them to maintain their function in the immunosuppressive environment. Subsequently, the study conducted a phase I clinical trial, and the results showed that some patients exhibited certain antitumor effects, although the durability of the response was limited [135]. Besides, prostate stem cell antigen (PSCA) is an antigen highly expressed on the surface of prostate cancer cells, and it is also an ideal target for CAR T-cell therapy. To investigate PSCA-targeted CAR T-cell therapy for metastatic castration-resistant prostate cancer, Dorff et al. conducted a phase I clinical trial. The results showed that some patients experienced tumor shrinkage and clinical improvement after treatment, although the durability of the effects was limited [136].

Overall, these studies provide increasing evidence for CAR T-cell therapy in prostate cancer, suggesting its potential, but further research is needed to enhance its efficacy and effectively manage side effects.

## 5. Conclusions

CD8^+^ T cells are a crucial component of the TME in PCa. Recent research highlights their significant roles in PCa development, metastasis, recurrence, and therapeutic resistance. One of the defining characteristics of the PCa TME is its immunosuppressive nature. In this milieu, the functions of CD8^+^ T cells are significantly suppressed, contributing to the immune evasion of PCa cells. Increasing CD8^+^ T-cell infiltration or reactivating/reversing the function of “exhausted” CD8^+^ T cells have been proven to be potent strategies for improving PCa treatment [113,116].

A range of therapeutic methods has been identified to significantly increase CD8^+^ T-cell infiltration, indicating the potential for combination with immunotherapy, such as immune checkpoint blockade (ICB), for PCa. Indeed, some therapeutic systems, including a prostate-specific four-antigen cassette expressing PSA, IRE, PAK4 inhibitors, and zoledronic acid-peptide vaccines, show a synergistic antitumor effect in PCa when combined with ICB due to the augmented CD8^+^ T-cell infiltration or reactivation of CD8^+^ T cells in tumors [119,122].

With the rapid advances in understanding the mechanisms of PCa immune escape, more specific targets for PCa treatment are likely to be identified. Firstly, uncovering new molecular signaling pathways associated with the function of “exhausted” CD8^+^ T cells is essential for developing new, specific targeting drugs to reactivate CD8^+^ T cells in PCa tumors. Secondly, the identification of new, specific tumor antigens is crucial for developing potent vaccines and CAR T-cell therapies. Additionally, given the limited efficacy of ICB in PCa, it is important to identify new immune inhibitory molecular signaling pathways or immune checkpoints involved in the immune escape of prostate cancer cells. Developing novel immune checkpoint blockade methods could enhance PCa treatment. Finally, gaining new insights into the interactions between CD8^+^ T cells, prostate cancer cells, and other stromal components will provide new targets for therapy.

## Figures and Tables

**Figure 1 biology-14-00544-f001:**
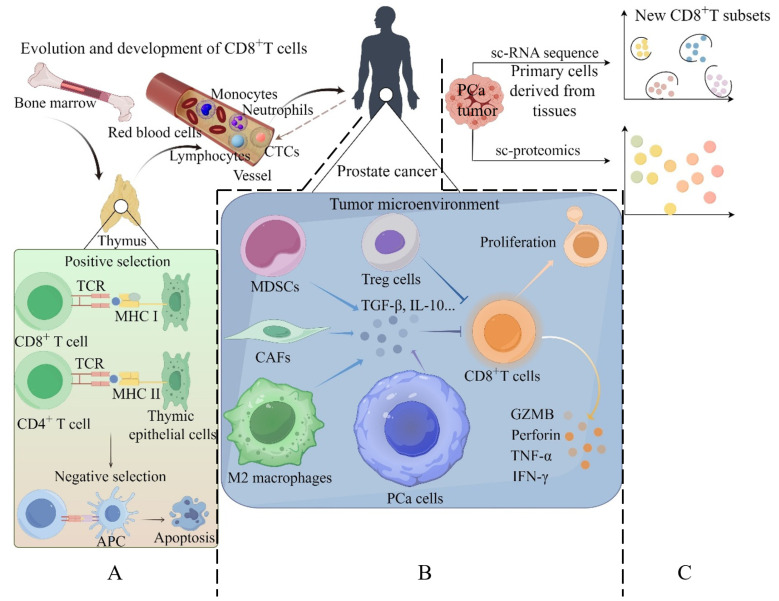
The role of CD8^+^ T cells in prostate cancer progression. CD8^+^ T cell precursors migrate from the bone marrow to the thymus for maturation and selection (**A**). Mature CD8^+^ T cells then infiltrate the prostate tumor site via the circulatory system. Within the tumor TME, these cells often become dysfunctional due to chronic stimulation by tumor antigens and various immunosuppressive factors (**B**). Advances in single-cell omics have uncovered novel CD8^+^ T cell subsets within tumors (**C**).

**Figure 2 biology-14-00544-f002:**
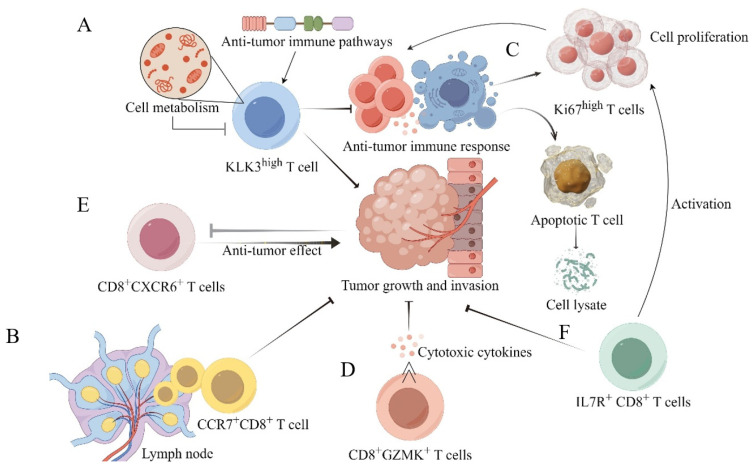
The function and role of newly identified CD8^+^ T subsets based on single-cell omics in the prostate cancer TME. Sc RNA sequence provides important insights into the immune cells within the PCa tumor microenvironment, revealing the heterogeneity of CD8^+^ T cells. Some new CD8^+^ T-cell subsets have been identified, and these subsets cooperate with each other in the TME (**A**–**F**). KLK3^high^ (Cluster 5) T cells exhibit low metabolic activity, suppressed antitumor immune pathways, and are associated with micro-metastases. Apoptotic (TC03) and Ki67^high^ (TC04) T cells are significantly enriched in prostate tumors compared to adjacent tissues and perform their respective functions [94]. CD8^+^CXCR6^+^ T cells, which act as effector cells, are markedly reduced and functionally inhibited by the tumor in malignant prostate cancer patients [92]. The CD8^+^GZMK^+^ (CTL-1) population represents an intermediate activation-to-exhaustion state within the tumor microenvironment and primarily executes antitumor functions. CCR7^+^CD8^+^ T cells are more prevalent in lymphatic metastases than in primary lesions, promoting tumor growth and metastatic dissemination, whereas IL7R^+^ CD8^+^ T cells are likewise enriched in lymphatic metastases and are associated with enhanced T-cell proliferation and suppression of tumor progression. Note: 
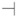
: suppress; >>: secrecte.

**Figure 3 biology-14-00544-f003:**
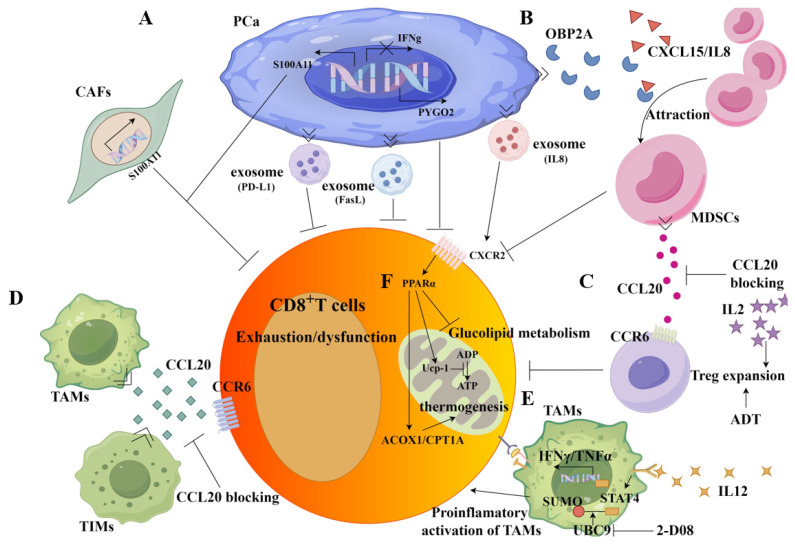
Various factors contribute to the dysfunction of CD8^+^ T cells in prostate cancer, and some strategies to reverse CD8^+^ T-cell dysfunction have been developed to enhance their therapeutic efficacy. Secreted proteins and exosomes from PCa cells contribute to the inhibition of antitumor T-cell responses by inducing CD8^+^ T-cell dysfunction, impairing glucose utilization, and promoting fatty acid catabolism. During the shrinking stage of PCa tumors after ADT, cancer cell-derived OBP2A interacts with CXCL15/IL8 to recruit MDSCs into the TME and a group of cytokines and chemokines, such as CCL20 and IL2, derived from MDSCs, TAMs, and Tregs, facilitate T-cell suppression and pro-tumor TME in PCa tissue. Targeted therapeutic strategies, including the use of CCL20-blocking antibody, can rescue CD8^+^ T-cell effector responses (**A**–**F**). Note: (1) 
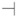
: suppress; >>: secrecte; 
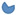
: OBP2A; 
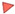
: CXCL15/IL8; 
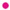
: CCL20 derived from MDSCs; 
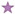
: IL2; 
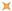
: IL12; 
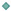
: CCL20 derived from TAMs; and 
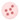
: exosome containing indicated proteins. (2) PCa: prostate cancer; CAFs: cancer-associated fibroblasts; TAMs: tumor-associated macrophages; MDSCs: myeloid-derived suppressor cells; and Tregs: regulatory T cells.

**Table 1 biology-14-00544-t001:** New CD8^+^ T subsets are identified based on single-cell omics.

CD8^+^ T Subset	The Role in PCa Progression	Reference
KLK3-high T-cell clusters (Cluster 5)	Its activity is significantly suppressed with the low level of cell metabolism and antitumor immune pathways, and is associated with micro-metastases of PCa.	[93]
TC03 subset and TC04 subset (apoptotic CD8^+^ T cells and Ki67^high^CD8^+^ T cells, respectively)	They are significantly enriched in prostate tumors compared to adjacent samples.	[94]
CD8^+^CXCR6^+^ T cells	It functions as effector T cells and its abundance is markedly declined in patients with malignant PCa.	[92]
CD8^+^GZMK^+^ T cells (CTL-1)	It is an intermediate state in the CD8^+^ T cell transition process from activation to exhaustion in PCa.	[95]
CCR7^+^CD8^+^ T cells	It is more common in lymphatic metastases compared with that in primary lesions, and promotes PCa metastasis and tumor growth.	[95]
IL7R^+^CD8^+^ T cells	It is more common in lymphatic metastases and is associated with T-cell proliferation and inhibition of tumor progression.	[95]

**Table 2 biology-14-00544-t002:** Summary of various therapeutic methods of targeting prostate cancer by CD8^+^ T cells.

Therapy Modality	Mechanism Overview	Advantages	Disadvantages
Immune Checkpoint Inhibitors (ICIs)	Monoclonal antibodies against PD-1/PD-L1 or CTLA-4 relieve tumor-mediated suppression of CD8^+^ T cells	- Established approvals in multiple cancers - Can reinvigorate existing tumor-specific T cells - Convenient intravenous dosing	- Generally low response rate as monotherapy in PCa - irAEs (e.g., dermatitis, colitis) - Limited predictive biomarkers
Cancer Vaccines	Delivery of prostate-cancer antigens (e.g., PAP, PSA, PSMA) to prime antigen-specific CD8⁺ T cells	- Good safety profile, low toxicity - Elicit de novo antigen-specific responses - Be amenable to combination with ICIs or chemotherapy	- Often weak immunogenicity alone, low objective response rates - Require adjuvants or multiple doses - Subject to patient immune tolerance
Adoptive Cell Therapy (CAR T/TCR T)	Ex vivo expansion and genetic engineering of patient CD8^+^ T cells (e.g., CAR-PSMA), then reinfusion	- Generate high-affinity, tumor-specific effector cells - Potential for long-term persistence and immunosurveillance - Some striking clinical responses	- High manufacturing cost and lengthy production - Risk of cytokine release syndrome and neurotoxicity - Solid-tumor microenvironment barriers
Cytokine Therapy (IL-2, IL-15, IL-12)	Systemic or localized delivery of cytokines to boost CD8^+^ T cell proliferation and functionality	- Potent systemic enhancement of T cell activity - Certain cytokines improve tumor infiltration - Synergize with vaccines or ICIs	- Severe systemic toxicities (e.g., capillary leak syndrome) - Limited efficacy as monotherapy - Require targeted delivery to reduce side effects
Bispecific T-cell Engagers (BiTEs)	Dual-binding antibodies that link CD3 on T cells with prostate tumor antigens (e.g., PSMA), activating cytotoxicity	- Mobilize endogenous CD8^+^ T cells without ex vivo manipulation - Flexible dosing - Early clinical signs of antitumor activity	- Short half-life often necessitating continuous infusion - Risk of cytokine release syndrome - Sensitivity to antigen heterogeneity
Antibody–Drug Conjugates (ADCs)	Tumor-antigen–targeted antibody delivers cytotoxic payload and promotes antigen release to activate CD8^+^ T cells	- Dual killing mechanisms: direct cytotoxicity plus immune activation - Focused toxicity at tumor site - Some ADCs in PCa have approvals/trials	- Limited T-cell activation when used alone; may require combination with immunotherapy - ADC-associated toxicities (e.g., thrombocytopenia, hepatotoxicity)
Combination Strategies	Rational combinations of the above (e.g., ICI + vaccine, CAR T + ICI, chemotherapy + ICI)	- Overcome resistance to single agents - Synergistically enhance CD8^+^ T-cell recruitment, activation, and persistence - Improved response rates in early studies	- Increased regimen complexity and overlapping toxicities - Optimal sequencing and dosing remain undefined - Higher cost and logistical challenges

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
