# Peer review of "Bolstering CD8+ T Cells’ Antitumor Immunity: A Promising Strategy to Improve the Response to Advanced Prostate Cancer Treatment"

_biology, 2025, doi:10.3390/biology14050544_

Round 1

Reviewer 1 Report

Comments and Suggestions for Authors

Review for Bolstering CD8+ T Cells Antitumor Immunity: a Promising Strategy to Improve the Response of Advanced Prostate Cancer Treatment

In this review, Dang et al discuss recent research and findings in the field of prostate cancer tumor heterogeneity. They also cover scRNA-seq and sc-proteomic studies which shed new light on tumor composition and CD8+ T-cell dynamics of prostate cancer. Additionally, the authors discuss various cellular and non cellular mechanisms that lead to CD8+ T-cell dysfunction. In the second half of the review, they highlight treatments and related clinical trials which aim to enhance CD8+ T-cells sensitivity in tumors to treat prostate cancer.

Overall the review is comprehensive, well-written and covers a wide range of research relating to prostate cancer and highlights how CD8+ T-cells can be harnessed to treat the tumor and its metastasis. I have a few suggestions that will help strenghten the manuscript:

Major Points: 

  • The figures are comprehensive and useful as reference. While reading the manuscript, however it appears that they are disjointed from the text and difficult to follow. To address this, the authors could cluster and label subpanels within the figures.
  • On a similar note, there are limited references in the text to figures and the table. As multiple topics are highlighted in the manuscript, authors could add additional figure references throughout the text to guide the reader.
  • Some information in the figures are not discussed in the text. For example, while the figures and legends include diagram of the thymus and maturation cycle of CD8+ T cells, the text lacks discussion of this process. 
  • As multiple prostate cancer treatments are highlighted in the paper, it would be useful to generate a chart similar to Table 1 summarizing the types of treatments that are available for prostate cancer targeting by CD8+ T-cells. If possible, the authors could include the pros and cons of each type of treatment to give the reader an overview of the field. 

Minor Points: 

  • Some stylistic changes may be useful, and specific sentences are verbose and can be difficult to follow. It may be useful to simplify them and proofread.
  • Some texts alternate between the use of abbreviations and the use of the full word (TME versus tumor microenvironment); since multiple abbreviations are used in the manuscript it would be useful to keep these consistent.
  • Line 28-31: “Adaptive immunity occurs when the body first encounters a specific pathogen and, through the activation and proliferation of immune cells, generates a targeted immune response, and it has memory, and its response is relatively slower, typically requiring several days, even weeks, to be fully activated.” The sentence is difficult to follow and may need proof-reading.
  • Line 56: Could use “tumors” instead of “tumor”
  • Some labels for Figure 1 overlap with the diagram images and may need to be readjusted as they can be difficult to read. 
  • The pink mass of cells in Figure 1 needs to be labeled, or it could be a separate panel. It is unclear if these are PCa cells.
  • Line 56-57: “For example, up-regulated nuclear cap-binding protein 2 (NCBP2) in PCa cells are closely associated with the decline in CD8+ T cells and the increase in Treg cells and neutrophils in tumor site, resulting an immunosuppressive microenvironment.” Could the authors expand on how NCBP2 leads to decline in CD8+ T cells and increase in T-regs?
  • Line 84-86: “The peripheral blood or nearby lymphe node usually are the source for the CD8+ T cells infiltrating into PCa tumor tissues. There is a correlation between peripheral blood and tumors.” In this sentence “lymphe” may be replaced with “lymph” and the sentences need to be refined because they are difficult to follow.
  • Line 103-104: “Dysfunction of CD8+ T cells is a key mechanism by which their anti-tumor effects are significantly impaired in PCa tissue, closely associated with PCa progression.” This sentence needs rephrasing, as it is difficult to understand.
  • Line 291: “the therapy involves collecting dendritic cells (DCs) from the patient’s blood, and then are exposed to a prostate cancer-related protein (PAP-GM-CSF) in the laboratory.” This sentence may be fixed for grammar
  • The authors may want to include a line or two introducing CAR-T cell therapies to orient the reader.
  • Line 334-336: “cells Thus, specific targeting YAP1 in ECM-CAF can switch the protumorigenic feature of CAFs to antitumor phenotypes in PCa. And when combined selective YAP1 depletion with anti-PD-1 antibodies can increase the immunotherapeutic effect.” This sentence has a missing period, “And” should not be capitalized and can be proofread.  

Author Response

Comments 1: The figures are comprehensive and useful as reference. While reading the manuscript, however it appears that they are disjointed from the text and difficult to follow. To address this, the authors could cluster and label subpanels within the figures.

Response 1: Thank you for pointing this out. I agree with this comment. We have clustered and labeled the subpanels in the figure with Arabic letters, and we added the details of these subpanels in the revised text.

Comments 2: On a similar note, there are limited references in the text to figures and the table. As multiple topics are highlighted in the manuscript, authors could add additional figure references throughout the text to guide the reader.

Response 2: Thank you for pointing this out. I agree with this comment. We have added additional figure references throughout the text highlighted in yellow font.

Comments 3: Some information in the figures are not discussed in the text. For example, while the figures and legends include diagram of the thymus and maturation cycle of CD8+ T cells, the text lacks discussion of this process.

Response 3: Thank you for pointing this out. I agree with this comment. We have added this discussion to lines 76–85 of the revised manuscript and highlighted it in yellow.

Comments 4: As multiple prostate cancer treatments are highlighted in the paper, it would be useful to generate a chart similar to Table 1 summarizing the types of treatments that are available for prostate cancer targeting by CD8+ T-cells. If possible, the authors could include the pros and cons of each type of treatment to give the reader an overview of the field.

Response 4: Thank you for pointing this out. I agree with this comment. For the chart you mentioned, we have created Table 2 and added it to the end of Section 4 in page 13, highlighting it in yellow. 

Comments 5: Some stylistic changes may be useful, and specific sentences are verbose and can be difficult to follow. It may be useful to simplify them and proofread.

Response 5: Thank you for pointing this out. I agree with this comment. Some of the sentences are verbose mainly because of the lengthy technical terms, and we simplified them in the revised manuscript. 

Comments 6: Some texts alternate between the use of abbreviations and the use of the full word (TME versus tumor microenvironment); since multiple abbreviations are used in the manuscript it would be useful to keep these consistent.

Response 6: Thank you for pointing this out. I agree with this comment. We have standardized the abbreviations and kept them consistent throughout the manuscript, highlighting in yellow in the revised manuscript.

Comments 7: Line 28-31: “Adaptive immunity occurs when the body first encounters a specific pathogen and, through the activation and proliferation of immune cells, generates a targeted immune response, and it has memory, and its response is relatively slower, typically requiring several days, even weeks, to be fully activated.” The sentence is difficult to follow and may need proof-reading.

Response 7:  Thank you for pointing this out. I agree with this comment. We apologized for the confusion. We have removed these sentences to make our manuscript easier to follow. 

Comments 8: Line 56: Could use “tumors” instead of “tumor”.

Response 8: Thank you for pointing this out. I agree with this comment. We have corrected it and highlighted it in yellow.

Comments 9: Some labels for Figure 1 overlap with the diagram images and may need to be readjusted as they can be difficult to read.

Response 9: Thank you for pointing this out. We readjusted the labels in Figure 1 to avoid overlapping..

Comments 10: The pink mass of cells in Figure 1 needs to be labeled, or it could be a separate panel. It is unclear if these are PCa cells.

Response 10: Thank you for pointing this out. I agree with this comment. We have labeled the pink mass of cells in Figure 1.

Comments 11: Line 56-57: “For example, up-regulated nuclear cap-binding protein 2 (NCBP2) in PCa cells are closely associated with the decline in CD8+ T cells and the increase in Treg cells and neutrophils in tumor site, resulting an immunosuppressive microenvironment.” Could the authors expand on how NCBP2 leads to decline in CD8+ T cells and increase in T-regs?

Response 11: Thank you for pointing this out. I agree with this comment. We have expanded the content followed as your suggestion in lines 46–51 of the manuscript and highlighted it in yellow.

Comments 12: Line 84-86: “The peripheral blood or nearby lymphe node usually are the source for the CD8+ T cells infiltrating into PCa tumor tissues. There is a correlation between peripheral blood and tumors.” In this sentence “lymphe” may be replaced with “lymph” and the sentences need to be refined because they are difficult to follow.

Response 12: Thank you for pointing this out. I agree with this comment. We have replaced the“lymphe”with“lymph”and revised this sentence in page 3,then highlighted it in yellow.

Comments 13: Line 103-104: “Dysfunction of CD8+ T cells is a key mechanism by which their anti-tumor effects are significantly impaired in PCa tissue, closely associated with PCa progression.” This sentence needs rephrasing, as it is difficult to understand.

Response 13: Thank you for pointing this out. I agree with this comment. We have revised this sentence and highlighted it in yellow in lines 109-112.

Comments 14: Line 291: “the therapy involves collecting dendritic cells (DCs) from the patient’s blood, and then are exposed to a prostate cancer-related protein (PAP-GM-CSF) in the laboratory.” This sentence may be fixed for grammar.

Response 14: Thank you for pointing this out. I agree with this comment. We have revised this sentence and highlighted it in yellow in lines 331-333.

Comments 15: The authors may want to include a line or two introducing CAR-T cell therapies to orient the reader.

Response 15: Thank you for pointing this out. I agree with this comment. We have added the content as your suggestion in lines 426–428 of the manuscript and highlighted it in yellow.

Comments 16: Line 334-336: “cells Thus, specific targeting YAP1 in ECM-CAF can switch the protumorigenic feature of CAFs to antitumor phenotypes in PCa. And when combined selective YAP1 depletion with anti-PD-1 antibodies can increase the immunotherapeutic effect.” This sentence has a missing period, “And” should not be capitalized and can be proofread.

Response 16: Thank you for pointing this out. I agree with this comment. We have revised this sentence and highlighted it in yellow lines 392-395.

Reviewer 2 Report

Comments and Suggestions for Authors

Dang et al. present their review of the key role of CD8+ T cells in prostate cancer, covering different aspects such as heterogeneity, dysfunction induced by the tumor microenvironment, and therapeutic strategies to boost their response. It particularly covers a topic with an increasing demand for review, such as the multiple subtypes being identified through “omics” technologies. The references used are appropriate and up-to-date and the Figures/Table presented reflect well the topics covered. Overall, the review is progressing in the right direction; however, it could be enhanced by considering the following remarks,

Introduction,

The introduction may benefit from a more organized structure. Certain aspects are redundant, and the events do not appear to follow a coherent order. The discussion on immunosuppression occurs multiple times, yet it could benefit from a more structured presentation. Additionally, focusing on the key aspects of tumor immunology related to prostate cancer is advisable to ensure it does not come across as a more general review of tumor immunology.

  • Line 21: Carefully review this paragraph about stromal cells; for instance, CD8+ T cells are not considered stroma cells; fibroblasts and endothelial cells are stroma cells.
  • Lines 27-34: Basic immunology information may be necessary; consider removing it.
  • Line 97 – 98: Consider revising this statement to avoid conveying an unintended impression that CD8+ T cells become effectors after direct activation with tumor antigens (unless you specifically refer to TCR-independent bystander activation). Perhaps by adding “tumor antigens recognized in the context of MHC class I molecules”.

Section 2

  • Consider changing the title of section 2 to exclude the word “dysfunction,” as this concept will be addressed in section 3.
  • Figure 1 and legend: Consider removing the section presenting bone-marrow precursors, thymus, and T cell ontology, since this process happens early in life, and it is unlikely to be connected to PCa, which frequently happens over 50. Additionally, the text written as “sc RNA sequence” and “sc proteomics” could be more clearly presented as “single-cell RNA sequencing” and “single-cell proteomics” to enhance clarity.
  • The title of section 2.2 may be “prognostic value of CD8+ T cell infiltration of prostate tumors.” This section presents controversial and thought-provoking data. Authors may explore and discuss differences between studies, variations in tumor staging, or the types of CD8+ T cells found in each case and attempt to reach a consensus for the section.
  • The concept of summarizing the various CD8+ T cell subsets identified through single-cell RNA sequencing and proteomics is intriguing and needed; however, the subtypes provided in the Table do not seem to have a counterpart in the text. The information about GZMK+ CD8 T cells is confusing: “transition process from activation to depletion in PCa.” Can the authors review what they mean by “depletion” in this sentence? Also, are TC03 and TC04 also CD8+ T cell subtypes? It is not clear either in the Table or in the text.
  • Line 169, perhaps the authors are referring here to the T cells’ cytotoxicity “score”?
  • Figure 2 might be a bit confusing, and the legend could use more explanation to help the reader. Table 1 fails to provide enough insight about each population to understand Figure 2. For example, in the figure, it looks like IL-7R+ cells are preventing tumor growth. However, Table 1 says otherwise. Consider reviewing Figure, Table and text; perhaps the Table might offer the most important aspect about each subtype found in each study.

Section 3,

Section 3 is well-written; however, the TAM that produces IL-12 in Figure 3 is unclear, and I did not find a reference to it in the text. The figure legend could be more self-explanatory, following the order of the figure so the reader can understand the figure by reading through the legend.

Section 4,

  • Line 290, please add a definition for mCRPC since is being first introduced in the manuscript.
  • Lines 307 – 310: This paragraph doesn’t seem to fit in the ICB section; perhaps a dedicated section to the microbiome and CD8 T cells in PCa is worth adding?
  • Line 311, consider perhaps using “therapeutic delivery systems” instead of “therapeutic systems.”
  • The ICB paragraph may benefit from some editing, perhaps adding some transition words when authors switch between ICB modalities.
  • Also, the ICB paragraph is missing important data on anti-PD-1/PD-L1; for instance https://www.cell.com/cancer-cell/fulltext/S1535-6108(23)00362-8
  • The chemotherapy paragraph could benefit from enrichment; combinations with immunotherapy could be included here.
  • Add “vaccines” to the 4.4 section title, as they are also presented in this section.

And as a final recommendation for the review in general, the authors might consider adding the species when referring to studies, as it is unclear whether they are referring to mouse or human data.

Comments on the Quality of English Language

English in general could be improved; the use of some transition words when changing subjects among paragraphs could enhance readability. 

Author Response

Comments 1: The introduction may benefit from a more organized structure. Certain aspects are redundant, and the events do not appear to follow a coherent order. The discussion on immunosuppression occurs multiple times, yet it could benefit from a more structured presentation. Additionally, focusing on the key aspects of tumor immunology related to prostate cancer is advisable to ensure it does not come across as a more general review of tumor immunology.

Response 1: Thank you for pointing this out. I agree with this comment. We have improved the structure of the introduction part by removing some redundant content followed as your suggestions. In this manuscript, we are focusing primarily on the immunosuppressive microenvironment of prostate cancer, especially the association of dysfunction of CD8+ T cell with the prostate cancer progression.

Comments 2: Line 21: Carefully review this paragraph about stromal cells; for instance, CD8+ T cells are not considered stroma cells; fibroblasts and endothelial cells are stroma cells.

Response 2: Thank you for pointing this out. I agree with this comment. We have revised the content you mentioned in lines 20–21 of the manuscript and highlighted it in yellow.

Comments 3: Lines 27-34: Basic immunology information may be not necessary; consider removing it.

Response 3: Thank you for pointing this out. We have removed this section in the revised manuscript.

Comments 4: Line 97–98: Consider revising this statement to avoid conveying an unintended impression that CD8+ T cells become effectors after direct activation with tumor antigens (unless you specifically refer to TCR-independent bystander activation). Perhaps by adding “tumor antigens recognized in the context of MHC class I molecules”.

Response 4: Thank you for pointing this out. I agree with this comment. We have revised the sentence according to your suggestions and highlighted them in yellow in lines 102–103.

Comments 5: Consider changing the title of section 2 to exclude the word “dysfunction,” as this concept will be addressed in section 3.

Response 5: Thank you for pointing this out. I agree with this comment. We have removed the word “dysfunction”.

Comments 6: Figure 1 and legend: Consider removing the section presenting bone-marrow precursors, thymus, and T cell ontology, since this process happens early in life, and it is unlikely to be connected to PCa, which frequently happens over 50. Additionally, the text written as “sc RNA sequence” and “sc proteomics” could be more clearly presented as “single-cell RNA sequencing” and “single-cell proteomics” to enhance clarity.

Response 6: Thank you for pointing this out. I agree with this comment. We apologized for the confusion. We added this content to Figure 1 to facilitate readers’ understanding and to align with certain parts of the text, so we want to keep it. We revised the “scRNAseq” into “sc RNA sequence” and “sc proteomics”followed as your suggestion.

Comments 7: The title of section 2.2 may be “prognostic value of CD8+ T cell infiltration of prostate tumors.” This section presents controversial and thought-provoking data. Authors may explore and discuss differences between studies, variations in tumor staging, or the types of CD8+ T cells found in each case and attempt to reach a consensus for the section.

Response 7: Thank you for pointing this out. I agree with this comment. We have revised the title of section 2.2 and added a discussion paragraph about it according to your suggestions and highlighted them in yellow in lines 122, 145-158.

Comments 8: The concept of summarizing the various CD8+ T cell subsets identified through single-cell RNA sequencing and proteomics is intriguing and needed; however, the subtypes provided in the Table do not seem to have a counterpart in the text. The information about GZMK+ CD8 T cells is confusing: “transition process from activation to depletion in PCa.” Can the authors review what they mean by “depletion” in this sentence? Also, are TC03 and TC04 also CD8+ T cell subtypes? It is not clear either in the Table or in the text.

Response 8: Thank you for pointing this out. I agree with this comment. We apologized for this confusion, the“depletion”in this sentence should be “exhaustion”. We have made the adjustions according to your suggestions and highlighted them in yellow in Table 2.

Comments 9: Line 169, perhaps the authors are referring here to the T cells’ cytotoxicity “score”?

Response 9: Thank you for pointing this out. I agree with this comment. We have made the correction according to your suggestions and highlighted them in yellow in line 194.

Comments 10: Figure 2 might be a bit confusing, and the legend could use more explanation to help the reader. Table 1 fails to provide enough insight about each population to understand Figure 2. For example, in the figure, it looks like IL-7R+ cells are preventing tumor growth. However, Table 1 says otherwise. Consider reviewing Figure, Table and text; perhaps the Table might offer the most important aspect about each subtype found in each study.

Response 10: Thank you for pointing this out. I agree with this comment. We have expanded the legend of Figure 2 and made the correction according to your suggestions and highlighted them in Figure 2 and Table 1 in yellow in line 209-218, 184-185.

Comments 11: Section 3 is well-written; however, the TAM that produces IL-12 in Figure 3 is unclear, and I did not find a reference to it in the text. The figure legend could be more self-explanatory, following the order of the figure so the reader can understand the figure by reading through the legend.

Response 11: Thank you for pointing this out. I agree with this comment. We have added the reference about the IL‑12 and TAM in Figure 3 in the revised manuscript. We have made the correction according to your suggestions and highlighted them in yellow in line 838-841.

Comments 12: Line 290, please add a definition for mCRPC since is being first introduced in the manuscript.

Response 12: Thank you for pointing this out. I agree with this comment. We have added a definition for mCRPC according to your suggestions and highlighted them in yellow in lines 327-330.

Comments 13: Lines 307 – 310: This paragraph doesn’t seem to fit in the ICB section; perhaps a dedicated section to the microbiome and CD8 T cells in PCa is worth adding?

Response 13: Thank you for pointing this out. I agree with this comment. We have added this part according to your suggestions and highlighted them in yellow in lines 344-362.

Comments 14: Line 311, consider perhaps using “therapeutic delivery systems” instead of “therapeutic systems.”

Response 14: Thank you for pointing this out. I agree with this comment. We have made adjustions according to your suggestions and highlighted them in yellow in line 365.

Comments 15: The ICB paragraph may benefit from some editing, perhaps adding some transition words when authors switch between ICB modalities.

Response 15: Thank you for pointing this out. I agree with this comment. We have added some transition words in this part according to your suggestions and highlighted them in yellow.

Comments 16: Also, the ICB paragraph is missing important data on anti-PD-1/PD-L1; for instance https://www.cell.com/cancer-cell/fulltext/S1535-6108(23)00362-8

Response 16: Thank you for pointing this out. I agree with this comment. we have added this part in lines 383-386,and the reference in lines 884-887.

Comments 17: The chemotherapy paragraph could benefit from enrichment; combinations with immunotherapy could be included here.

Response 17: Thank you for pointing this out. I agree with this comment. We have added this part according to your suggestions and highlighted them in yellow in lines 418-423.

Comments 18: Add “vaccines” to the 4.4 section title, as they are also presented in this section.

Response 18: Thank you for pointing this out. I agree with this comment. We have added “vaccines” according to your suggestions and highlighted them in yellow in line 425.

Comments 19: And as a final recommendation for the review in general, the authors might consider adding the species when referring to studies, as it is unclear whether they are referring to mouse or human data.

Response 19: Thank you for pointing this out. I agree with this comment. We have added the information of the species (human or mouse) in yellow through the manuscript.

Reviewer 3 Report

Comments and Suggestions for Authors

In this manuscript, Dang et al. provided a quite comprehensive review of the roles of CD8⁺ T cells in treating advanced prostate cancer. The authors discussed CD8+ T cells’ behavior in tumor microenvironment, the mechanisms underlying the suppression of immunity and emerging immunotherapeutic strategies to restore their antitumor activity. The authors synthesize findings from numerous up-to-date publications covering new omics data, preclinical models, and clinical studies. The figures are well prepared and illustrative. 

However, there are some minor issues that would need to be addressed before publication. 

  1. There is controversy on whether the effect of CD8+ T cell infiltration in PCa is good or bad. Please cite manuscripts on both sides of the viewpoint and briefly discuss the reason behind these conflicts, and identify knowledge gaps for future researchers.
  2. The authors did a great job in summarizing current clinical outcomes in Section 4. However, given the depth of the topics covered, it would bring much clarity if clinical results and ongoing clinical trials are summarized in an additional table. 

Author Response

Comments 1: There is controversy on whether the effect of CD8+ T cell infiltration in PCa is good or bad. Please cite manuscripts on both sides of the viewpoint and briefly discuss the reason behind these conflicts, and identify knowledge gaps for future researchers.

Response 1: Thank you for pointing this out. I agree with this comment. We have cited references about both sides in this part, and we have added a discussion paragraph about it according to your suggestions and highlighted them in yellow in lines 145-158.

Comments 2: The authors did a great job in summarizing current clinical outcomes in Section 4. However, given the depth of the topics covered, it would bring much clarity if clinical results and ongoing clinical trials are summarized in an additional table.

Response 2: Thank you for pointing this out. I agree with this comment. In line with suggestions from both Reviewer 1 and Reviewer 3, we have added Table 2 at the end of Section 4 and highlighted it in yellow in page 13.